# The Biological Function of MicroRNAs in Bone Tumors

**DOI:** 10.3390/ijms23042348

**Published:** 2022-02-21

**Authors:** Sarah Adriana Scuderi, Giovanna Calabrese, Irene Paterniti, Michela Campolo, Marika Lanza, Anna Paola Capra, Luca Pantaleo, Stefania Munaò, Lorenzo Colarossi, Stefano Forte, Salvatore Cuzzocrea, Emanuela Esposito

**Affiliations:** 1Department of Chemical, Biological, Pharmaceutical and Environmental Sciences, University of Messina, Viale Ferdinando Stagno D’Alcontres, 98100 Messina, Italy; sarahadriana.scuderi@unime.it (S.A.S.); giovanna.calabrese@unime.it (G.C.); ipaterniti@unime.it (I.P.); michela.campolo@unime.it (M.C.); marika.lanza@unime.it (M.L.); lucapantaleo2010@libero.it (L.P.); salvator@unime.it (S.C.); 2Department of Clinical and Experimental Medicine, University of Messina, Viale Ferdinando Stagno D’Alcontres, 98100 Messina, Italy; annapaola.capra@unime.it; 3Istituto Oncologico del Mediterraneo, Via Penninazzo 7, 95029 Viagrande, Italy; stefania.munao@grupposamed.com (S.M.); lorenzo.colarossi@grupposamed.com (L.C.); 4IOM Ricerca Srl, Via Penninazzo 11, 95029 Viagrande, Italy; stefano.forte@grupposamed.com

**Keywords:** microRNA, bone tumor, osteosarcoma, Ewing’s sarcoma, chondrosarcoma, biomarker

## Abstract

Micro ribonucleic acids (miRNAs) are small endogenous noncoding RNAs molecules that regulate gene expression post-transcriptionally. A single miRNA is able to target hundreds of specific messenger RNA (mRNAs) by binding to the 3′-untranslated regions. miRNAs regulate different biological processes such as cell proliferation, differentiation and apoptosis. Altered miRNA expression is certainly related to the development of the most common human diseases, including tumors. Osteosarcoma (OS), Ewing’s Sarcoma (ES), and Chondrosarcoma (CS) are the most common primary bone tumors which affect mainly children and adolescents. A significant dysregulation of miRNA expression, in particular of mir-34, mir-21, mir-106, mir-143, and miR-100, has been revealed in OS, ES and CS. In this context, miRNAs can act as either tumor suppressor genes or oncogenes, contributing to the initiation and progression of bone tumors. The in-depth study of these small molecules can thus help to better understand their biological functions in bone tumors. Therefore, this review aims to examine the potential role of miRNAs in bone tumors, especially OS, ES and CS, and to suggest their possible use as potential therapeutic targets for the treatment of bone tumors and as biomarkers for early diagnosis.

## 1. Introduction

MicroRNAs (miRNAs) are short noncoding RNAs, consisting of 18–25 nucleotides, which regulate post-transcriptional silencing of genes [1,2]. MiRNAs are able to control many cellular processes, including growth, proliferation, differentiation, and cell death [3]. A single miRNA can target multiple genes; therefore, it is important to understand the biological function of each miRNA [4]. miRNAs are localized as clusters in the genome, and can be classified into homo-clusters (miRNAs belonging to the same family) and hetero-clusters (miRNA genes from different families as based on nucleotide sequence homology) [5,6]. A single miRNA cluster contains two or more miRNA genes transcribed from adjacent nucleotide sequences in the same orientation. MiRNAs regulate different cell signaling pathways by targeting secondary messengers and transcription factors (TFs). Because members of miRNA clusters and TFs can regulate each other, cross-talk between them is important for maintaining cellular homeostasis. A dysregulation of miRNA expression may cause alterations in various biological processes, which contributes to the pathogenesis of many human diseases, including tumors [7]. Many studies have revealed that miRNAs can act as tumor suppressor genes or oncogenes, contributing to the initiation and progression of malignancy [4,8].

Altered miRNA expression profiles in primary bone tumors have mostly been identified as Osteosarcoma (OS), Ewing’s sarcoma (ES), and Chondrosarcoma (CS) [9]. Primary bone tumors include a wide range of histotypes with variable clinical outcomes and poor prognoses, many of which are rare. They arise in the metaphysis and diaphysis of long bones, and extend secondarily into the epiphysis [10].

OS, ES and CS etiology remains unknown, and conventional treatments include generally neoadjuvant chemotherapy, surgery, and sometimes irradiation [11,12]. Many studies have demonstrated that miRNAs can influence progression, invasion, and metastasis in bone tumors and negatively affect chemotherapy response; thus, their regulation could be beneficial to prevent or arrest disease progression [3,13,14]. As miRNA levels are stable in the blood and each type of tumor has a distinct miRNA signatures [15], miRNAs can be used as biomarkers for the early diagnosis and therapy of bone tumors [16].

Therefore, considering their critical regulatory role in multiple biological processes and diseases including tumors, in this review we focus on the function and potential role of miRNAs in bone tumors, particularly in OS, ES and CS.

## 2. Biogenesis and Biological Function of miRNAs

Following the discovery of the first miRNA in *Caenorhabditis elegans* by Lee et al. [17] in 1993 scientific interest in miRNAs has increased, resulting in the discovery of hundreds of miRNAs in humans. miRNAs are small, single-stranded, non-coding RNAs of about 18–24 nucleotides and are involved in many biological processes, including the proliferation, differentiation, and apoptosis of cells [2,17,18,19,20]. Their biogenesis is a complex multi-step process that includes nuclear and cytoplasmic synthesis and requires the involvement of different enzymes (Figure 1). They are mainly transcribed by RNA polymerase II as longer transcripts (called primary miRNAs or pri-miRNAs [21]) and processed in the nucleus by the heterotrimeric microprocessor complex, which comprises the RNase III enzyme Drosha and the protein DiGeorge syndrome critical region gene 8 (DGCR8), into precursor miRNAs of about 70–80 nucleotides (pre-miRNAs) [5,22]. Successively, exportin-5 protein, a cytoplasmic transporter, transfers pre-miRNAs from the nucleus into the cytoplasm. In the cytoplasm, the RNase III enzyme Dicer binds pre-miRNAs and cleaves them into mature miRNAs duplexes of ~22 nucleotides long. After strand separation, single-stranded mature microRNAs are bound by the Argonauta protein (Ago) and incorporated into the enzymatic complex RISC (RNA-induced silencing complex) to exert their biological function [23].

MiRNAs can be secreted into extracellular fluids and transported to target cells via vesicles such as exosomes or by binding to proteins such as Argonautes [24,25]. Mature miRNAs regulate protein production by binding to complementary target mRNAs via the RISC complex [26].

miRNAs generally bind to their target mRNAs via complementary base-paring between the miRNA seed region and sequences within mRNA 3′-untranslated regions (3′UTRs) [6].

Sequence complementarity between the microRNA and its target is preferentially located at the 5′ end of the microRNA, termed the seed, which consists of 2–8 nucleotides [27]. The interaction of miRNAs with their target genes depends on several factors, including the subcellular location of the miRNAs, the excess or lack of miRNAs and target mRNAs, and the respective affinities of miRNA–mRNA [23].

Each miRNA is capable of regulating up to a hundred or more mRNA species, generating an intermolecular network. The biogenesis and maturation of miRNAs are partially regulated by several cellular mechanisms, including deoxyribonucleic acid (DNA) methylation and histone deacetylation [28,29]. Furthermore, a single nucleotide change on a primary miRNA can greatly influence its stability and maturation or alter its activity. miRNAs are able to regulate gene expression at the post-transcriptional level by binding to target mRNAs, causing translation inhibition and consequent protein synthesis inhibition [30]. Studies have revealed that miRNAs regulate up to 30% of the protein-coding genes in the human genome; approximately 52% of human miRNAs are located in intergenic regions, 40% lie within intronic regions of genes, and the final 8% are exonics [31,32]. miRNAs play a key role in the maintenance of cellular homeostasis and their dysregulation is often associated with human diseases such as diabetes, cardiovascular disease, and particularly tumors [33,34,35]. Many human miRNAs and their target sequences exhibit considerable sequence homology across different species, making them good candidates for study in animal models [36].

Therefore, miRNAs represent very interesting molecules to study for their ability to regulate several pathophysiological processes as well as for their possible use as therapeutic targets in human diseases.

## 3. Role of miRNAs in Osteogenesis and Bone Homeostasis

Bone tissue is continually remodeled throughout the lifetime of an individual, and includes four specific cell types: osteocytes, osteoblasts, osteoclasts in bone, and chondrocytes in cartilage. Bone homeostasis is preserved through the dynamic balance between osteoclastic bone resorption and osteoblastic bone formation. miRNAs are important regulators of the bone resorbing activity maintained by osteoclasts as well as of the osteoblast proliferation and differentiation processes (Figure 2) [37].

The dysregulation of miRNAs is an important pathological factor in bone degeneration and resorption as well as in bone-related diseases such as tumors [38]. miRNAs regulate osteogenic differentiation and bone formation via such crucial signaling pathways as the transforming growth factor-beta (TGF-β)/bone morphogenic protein (BMP), the Wingless/Int-1(Wnt)/β-catenin, and Notch [39]. An increasing number of miRNAs have been shown to positively regulate osteoblast differentiation and bone formation by targeting regulators of osteogenesis or to negatively regulate it by targeting important osteogenic factors; miR-29b promotes osteoblastic differentiation by down-regulation of many Wnt signaling inhibitors, and likely contributes to this effect by promoting a positive feedback loop [40], while miR-141 and miR-200a inhibit osteoblast differentiation through repression of the osteogenic transcription factor DLX5. MicroRNA-141 and -200a are involved in bone morphogenetic protein-2-induced mouse pre-osteoblast differentiation by targeting distal-less homeobox 5 [41], miR-182 inhibits osteoblast proliferation and differentiation by targeting and repressing the Foxo1 gene [42], and miR-34c alters osteoblast differentiation and function by suppressing Notch signaling through the inhibition of Notch1, Notch2, and Jag1 [43].

In addition, chondrocytes, which are essential regulators of longitudinal bone growth, have been shown to be regulated by various miRNAs such as miR-140, a positive regulator of chondrogenesis that contributes to craniofacial development and endochondral bone formation through the inhibition of HDAC4 [44]. Conversely, miR-199a and miR-145 have been shown to negatively regulate chondrocyte differentiation [45]; miR-199a downregulates the expression of SMAD1 protein [46], whereas miR-145 targets SOX9, a key transcription factor involved in chondrogenesis and cartilage formation [45,46]. Similar miRNA-based regulation has been observed during osteoblast differentiation and in bone resorbing cells. Suppression of miR-21 has been associated with upregulation of osteoclast suppressor programmed cell death protein 4 (PDCD4) and downregulation of osteoclast marker cathepsin K (CTSK), suggesting that miR-21 may be involved in bone biology both by promoting mobilization of osteoblast precursors and by regulation of osteoclast survival and differentiation [47].

## 4. Relationship between miRNAs and Cancers: Focus on Bone Tumors

Several studies have demonstrated that miRNAs both regulate biological processes such as cellular proliferation, differentiation and apoptosis and modulate different pathological conditions. Specifically, numerous miRNAs have been identified and characterized in many human diseases [28], including tumors [15,48]. The term “tumor” indicates a group of pathologies characterized by uncontrolled cell replication and spread that creates an abnormal mass of tissue with altered functionality; tumors represent some of the most frequent and aggressive human pathologies.

The first evidence of the involvement of miRNAs in human tumors was published in 2002 by Calin et al. [49]. In this study, the authors revealed that miR-15 and miR-16 at chromosome 13q14 are deleted or downregulated in the majority of chronic lymphocytic leukemia (CLL) cases, inducing apoptosis by direct suppression of Bcl-2 (B cell lymphoma 2) in CLL cells.

Many studies have focused on the biological functions of miRNAs, including their effects on tumorigenesis and in particular on the progression, apoptosis, and proliferation of tumor cells. Genetic or epigenetic alterations, dysregulation of TFs, and abnormal miRNA biogenesis can alter miRNA expression, contributing to the onset and progression of tumors [15,28,50]. Many human miRNA genes are thought to be located in cancer-associated regions or at fragile sites of chromosomes which are prone to deletion, amplification, and mutations in cancer cells [4]. The dysregulation of miRNA expression can influence tumorigenesis if mRNA targets are encoded by tumor suppressor genes or oncogenes [8,51,52,53,54]. Therefore, altered miRNA expression may cause several complications in tumor progression [29]. It has been demonstrated that neo-angiogenesis is a crucial step during metastatic processes, allowing cells to disseminate through systemic circulation; in this context, miRNAs are able to control tumor progression through the process of angiogenesis, promoting or inhibiting the proliferation of endothelial cells [55]. miRNAs can act as oncogenes, as tumor suppressors, or in some cases as both, in accordance with their expression and their role in malignancies [56]; moreover, considering their different roles, miRNAs can affect the cell differentiation and apoptosis processes in tumor [57,58,59,60,61,62].

The main miRNAs involved in tumor are described below and shown in Table 1.

miR-21: significantly overexpressed in different tumor types, in particular in bone tumors [63], it is involved in the regulation of many tumor suppressor genes and apoptosis-related proteins including tropomyosin-1 (*TPM1*), programmed death protein 4 (*PDCD4*), and metalloproteinase inhibitor 3 (*TIMP3*) [64].

miR-34: comprising miR-34a, miR-34b and miR-34c, it is encoded by two different transcriptional units: miR-34a is located at chromosome 1p36.22 and has a unique transcript, while miR-34b and miR-34c hold one transcript in common, located at chromosome 11q23.1 [65]. miR-34 was the first miRNA shown to be directly regulated by the tumor suppressor p53 [66].

Let-7: acts as a tumor suppressor by controlling metastasis and cell proliferation; its over-expression is related to cell growth inhibition and reduction in cell cycle progression [67,68].

miR-106a/b: firstly identified as an oncogenic factor in a variety of tumors [69], miR-106b is overexpressed in bone tumors and its expression is related to lung metastasis and clinical tumor stages [70].

miR-125: acts as a tumor suppressor in cancer; miR-125b is frequently down-regulated in osteosarcoma and its ectopic restoration suppresses cell proliferation and migration, indicating that signal transducer and activator of transcription 3 (STAT3) is its direct and functional target [71].

miR-143: a tumor-suppressing miRNA, its down-regulation has been reported in several human cancers. Sun et al. [72] revealed that the down-regulation of miR-143 is closely related to the development of bone tumors and that its main target is represented by Fos-related antigen 2 (FOSL2), which plays a key role in bone development.

miR-208a: Fa et al., displayed that miR-208a increases the viability, migration, and clonogenicity of osteosarcoma cells via downregulation of PDCD4 and activation of the ERK1/2 pathway, highlighting its key role in bone tumor pathogenesis [73].

**Table 1 ijms-23-02348-t001:** List of microRNAs involved in tumor development. The table reports the prevalent bone tumors and target genes as well as their respective functions.

MicroRNA	Prevalent Tumor	Target Gene	Function	References
miR-21	OS; ES;	*PTEN*, *TPM1*, *PDCD4*	Increases proliferation	[63,64]
miR-34	OS; ES;	*P53*; *Notch*	Inhibits proliferation	[65,66]
Let-7a	OS; ES;	*HMGA2*; *E2F2*	Reduces growth	[67,68,74]
miR-106	OS; ES;	*PTEN*; *CDKN1A*;	Increases proliferation	[45]
miR-125b	OS; ES;	*P53*; *Bak*; *STAT3*	Inhibits proliferation	[75]
miR-143	OS; ES;	*FOSL2*; *TFF3*;	Inhibits proliferation	[72]
miR-100	CS;	*mTOR*; *IGFIR*;	Inhibits proliferation	[76]
miR-16	OS;	*RAB23*; *SMAD3*;	Inhibits proliferation	[77]
miR-30a	CS;	*SOX4*; *Notch1*;	Reduces proliferation	[59]
miR-181a	CS;	*RGS16*; *RASSF1A*;	Increases proliferation	[78]
miR-19a	OS;	*SOCS6*; *PTEN*;	Increases proliferation	[79]
miR-199a	OS;	*PIAS3*, *Smad1*	Reduces proliferation	[59]
miR-145	CS; OS;	*SOX9*; *MMP16*	Inhibits invasion	[58]
miR-140	OS; CS;	*HDAC4*; *p21*;	Reduces proliferation	[57]
miR-208a	OS;	*PDCD4*; *ERK1/2*;	Increases proliferation	[73]
miR-221	OS;	*PTEN*;	Increases proliferation	[70]
miR-22	OS;	*ACLY*, *TWIST1*;	Reduces proliferation	[52]
miR-101	OS;	*ROCK1*; *EZH2/Wnt/β-Catenin*	Reduces proliferation and invasion	[53]
miR-483	OS;	*STAT3*	Increases proliferation	[54]
miR-191	OS,	*BDNF*; *CDK6*;	Increases proliferation	[80]
miR-99a	OS;	*TNFAIP8*	Reduces proliferation	[61]
miR-27a	OS	*TET1*	Increases proliferation	[60]
miR-424	OS;	*CcnA2*	Reduces proliferation	[62]
miR-449a	OS;	*CcnA2*	Reduces proliferation	[62]
miR-30c	OS;	*MEF2D*	Reduces proliferation	[59]
miR-206	OS;	*HDAC4*	Increases proliferation	[58]
miR-665	OS,	*Rab23*	Reduces metastasis and invasion	[81]
miR-15a	OS;	*Bcl2*	Reduces metastasis and invasion	[82]
miR-17-5p	OS;	*BRCC2*	Increases proliferation	[83]
miR-590-5p	OS;	*KLF5*	Reduces proliferation	[84]
miR-212	OS;	*SOX4*	Reduces proliferation	[85]
miR-132	OS;	*SOX4*	Reduces proliferation	[86]
miR-23b	OS;	*PI3K/AKT*	Increases proliferation	[87]
miR-183	OS;	*Ezrin*	Reduces metastasis	[88]

Over the last decade great attention has been given to the role of miRNAs in bone tumors [66,78,89]. Primary malignant bone tumors represent approximately 6% of all childhood malignancies, of which OS, ES, and CS are the most common tumors.

Chromosomal translocations play a key role in the pathogenesis of bone tumors such as OS, ES, and CS. There are three substantial molecular mechanisms by which chromosomal translocations cause tumorigenesis: (i) formation of a chimeric gene, in which the fusion transcript acts as an aberrant transcription factor causing transcriptional deregulation, as happens in ES; (ii) promoter swap, in which a gene promoter that is normally highly expressed in bone is fused to the coding sequence of another gene, causing its up-regulation and consequent altered signaling; and (iii) disruption of a specific gene, causing inactivation or altered function of the gene, as happens in OS [90].

In this context, many studies have reported that specific miRNAs are involved in the process of bone development and differentiation (Figure 3) [26,91].

Bone mass maintenance depends mainly on the balance between osteoblast-mediated bone formation and osteoclast-mediated bone resorption, which in turn are regulated by miRNAs such as miR-143 [72]. Thus, miRNAs are one of the most important modulators in bone remodeling, suggesting that abnormal miRNA expression is related to the development and progression of primary bone tumors (Table 2) [9].

Therefore, a great deal of research has focused on the role and expression of specific miRNAs in bone tumors, in particular in OS, ES, and CS (Figure 4), seeking to evaluate their possible use as new therapeutic targets and alternative diagnostic tools [66,78].

### 4.1. Role of miRNAs in Osteosarcoma

OS is one of the most common malignant bone tumors to affect mainly children and adolescents, with an incidence of approximately 4–5 cases per million, and is characterized by very intense and progressive pain [92]. Several studies have suggested that OS arises from primitive mesenchymal bone-forming cells that undergo aberrant alterations in the differentiation process [11,75]. Tall stature and male sex may be risk factors for OS, although these findings are drawn from a limited number of studies and remain controversial [11]. OS is characterized by vast genomic instability and by multiple genomic aberrations, which are detected in the majority of OS cases (58%) [92]. These chromosome abnormalities are caused by errors in mitosis, germline mutations, deletions, duplications, and unbalanced translocations and can include gain of chromosome 1, loss of chromosomes 9, 10, 13, and/or 17 and partial or complete loss of the long arm of chromosome 6. Development of malignant osteosarcoma cells is thought to be related to alteration of many signaling pathways due to genetic mutations, many of which are at least partially regulated by miRNAs. Emerging evidences has shown that alteration of miRNAs, in particular of miR-21, is involved in OS development and progression [63,93]. Zhang et al. [94] revealed that miR-21 is highly expressed in OS tissues, suggesting that it plays a key role in tumor cell proliferation, migration, invasion, and apoptosis. It has been documented that miR-21, located on human chromosome 17q23.2, acts as an oncogenic miRNA. miR-21 is able to negatively regulate phosphatase and tensin homologue deleted on chromosome ten (PTEN), tropomyosin 1 (TPM1), and apoptosis protein 4 (PDCD4), which in turn induces tumor cell growth, migration, invasion, and metastasis [95]. Studies have shown that inhibition of miR-21 expression can therefore delay progression of OS, proving it to be a potential therapeutic target and suggesting its possible use as a biomarker for OS [89].

Another important study conducted by Ha et al. [96] revealed that miR-34 cluster is involved in OS development as well. The miR-34 cluster consists of mir-34a, miR-34b, and miR-34c and affects expression of target genes, in particular the human p53 gene, which encodes for the p53 tumor suppressor protein and plays a key role in maintaining genomic stability. This study revealed that miR-34a inhibited osteosarcoma cell proliferation and metastasis progress thanks to its direct action on the p53 gene [96,97]. Moreover, miR-34 demonstrated the ability to inhibit cancer growth by inactivating the Notch signaling pathway, which is involved in different cellular processes such as differentiation, proliferation, migration, and angiogenesis [74].

Although the correlation between the expression of miRNAs and the clinicopathological signs of OS remains unclear, previous studies have found that the levels of miR-221 and miRNA-106a were elevated in osteosarcoma patients compared with normal bone cells, suggesting that the miRNA levels are correlated with tumor stage and degree of differentiation [70]. The interaction of miR-221 and mRNA-3′UTR with the tumor suppressor PTEN attenuates the ability of PTEN to control cell proliferation, apoptosis, metastasis, and invasion, which, in turn upregulates the expression of cyclin-dependent kinase inhibitor p27kip1, promoting the growth of OS [98].

Moreover, Sun et al. [72] investigated the relationship in OS between miR-143 and FOSL2, a key regulator of bone development. Because overexpression of FOSL2 leads to OS development [98], their study showed that miR-143 expression was lower in OS tissues compared with normal bone tissues, and that miR-143 was able to inhibit the proliferation, migration, and invasion of OS by reducing the expression of FOSL2.

Jonee et al. [99], in an interesting study conducted on patient OS samples, revealed that miRNA expression reflects the pathogenesis and malignant degree of OS. This study demonstrated that patients affected by OS had high expression of miR-181a, miR-181b, and miR-181c and reduced expression of miR-16, miR-29b, and miR-142-5p [100]. Recently, many studies have focused on the roles of miR-16 and miR-665 in OS. Specifically, it has been demonstrated that miR-16 and miR-665 are down-regulated in OS cell lines (MG63, U2OS) and OS samples [77,81] and that both were able to suppress tumor cell progression and invasion through RAB23 modulation, a protein encoded by the RAB23 gene which is involved in tumor proliferation [81,101].

miR-19, an oncogenic component of the polycistronic miR-17~92 cluster, plays a key role in OS development as well. Sun et al. [79] demonstrated that miR-19 was up-regulated in OS by modulation of SOCS6, a member of the SOCS-suppressor family involved in tumorigenesis. Specifically, the authors demonstrated that SOCS6 expression was negatively correlated with miR-19 in OS tissue, confirming its central role in OS progression.

However, despite the many studies on the role of miRNAs in OS, more in-depth knowledge is needed in order to provide novel and effective therapeutic approaches to tumor treatment.

### 4.2. Role of miRNAs in Ewing’s Sarcoma

ES is the second most common and aggressive bone tumor to mainly affect children and young adults. ES begins between the ages of 5 and 30, with an annual incidence estimated at 1/312,500 children under the age of 15 [12,65]. ES is characterized by chromosomal translocation involving fusion of the 5’segment of the Ewing’s sarcoma breakpoint region 1 (EWS) gene to the 3’segment of an E26 transformation-specific (ETS) family gene with friend leukemia integration 1 transcription factor (FLI1) to generate EWS-FLI1 protein. The clinical presentation of this tumor is varied and includes mostly pain, fever, and weight loss. Delay in diagnosis is more common in patients with ES than in OS, as the tumor is often not clinically evident until it reaches an appreciable size.

The EWS-FLI1 fusion protein represents a key oncogenic event in ES and is responsible for the transcriptional deregulation of several genes involved in ES pathogenesis [12]. Despite the mechanisms underlying the development of ES not yet being fully understood, several studies have suggested that the EWS/FLI1 protein induces important changes in miRNA expression, mostly in miR-21 and miR-145 [14,102], contributing to disease progression.

Recent reports have demonstrated that uncontrolled apoptosis is an important feature of ES, suggesting the involvement of the Bcl-2 pathway in ES pathophysiology [12,103]. Bcl-2 is an anti-apoptotic protein which, under normal conditions, maintains mitochondrial integrity and cell survival; in this context, many studies have focused on miR-21 functions in ES. Mir-21 acts as an oncogene and is able to modulate tumorigenesis through regulation of the Bcl-2 gene, showing that Bcl-2 up-regulation may be caused by miR-21 over-expression preventing tumor-cell apoptosis [76].

In the last decade, 58 different miRNAs have been identified in ES patients, including miR-21, miR-30b, miR-27a, miR-106b, miR-181a/b, miR-130a, let-7e, let-7b, let-7f, let-7g, let-7a, and miR-34a, all of which are up-regulated [103,104]. In particular, let-7a is a direct EWSFLI-1 target which plays a key role in ES pathogenesis; let-7a acts as tumor suppressors thanks to its ability to silence numerous genes that encode oncogenic proteins (such as high-mobility group AT-hook (HMGA2)*,* a DNA binding protein highly expressed in cancer cells), resulting in a reduction of tumor growth [105].

Attention has been given to the role of miR-34a and miR-34b in ES as well, and it has been demonstrated that miR-34a and miR-34b play a key role in ES development by acting on the Notch gene [65]. An interesting aspect of Notch signaling is that it can act as an oncogene or as a tumor suppressor. Studies have demonstrated that the suppression of EWS-FLI1 reactivates Notch signaling in ES cells, resulting in cell cycle arrest and apoptosis [106]. The miR-34a/Notch axis has been widely studied for its role in malignant diseases, including ES, suggesting that both miR-34a and miR-34b potentially have an oncogenic role in ES via downregulating of Notch [94].

The role of miR-125b in ES has been investigated as well. Mir-125b acts as a tumor suppressor in several types of tumors, including breast cancer and hepatocellular carcinoma; in contrast, miR-125b acts as an oncogene in ES by targeting p53 and Bak, a pro-apoptotic mitochondrial protein. The inactivation of p53 during the development and progression of ES has been explained through the interaction between p53 and the EWS/FLI1 gene. Therefore, according to this evidence, miR-125b could represent a novel diagnostic biomarker in ES thanks its ability to regulate p53 and Bak expression [75,107].

Kavano et al. [108] investigated the role of miR-181c in ES cell lines, showing that miR-181c was significantly up-regulated. In this context, the use of anti-miR-181c in ES cell lines was proposed; this caused greater expression of FAS2, a signaling pathway that plays a key role in the physiological regulation of apoptosis. Thus, anti-miR181c demonstrated a reduction in cell growth and promoted apoptosis in ES cell lines via the FAS2 pathway, suggesting that unregulated expression of miR-181c contribute to ES progression.

In addition, miR-30a-5p plays a key role in ES physiopathology. Duval et al. [109] demonstrated that miRNAs, in particular miR-30a-5p, may be involved in the regulation of CD99 protein by EWS-FLI1. CD99 is a ubiquitous trans-membrane glycoprotein implicated in several cellular processes including cell adhesion, migration, and apoptosis [110]. In vitro and in vivo studies demonstrated that miRNA-30a-5p was down-regulated in ES through CD99 modulation, suggesting post-transcriptional regulation and a tumor-suppressing effect [91,109,111].

McKinsey et al. [104] focused on the role of miR-27a in ES. MiR-27a negatively regulates insulin-like growth factor-1 (IGF-1) expression in ES. Generally, IGF-1 stimulates cell growth, proliferation, and differentiation; moreover, it is involved in tumor development [112]. In this context, miR-27 repressed the IGF-1 signaling pathway, a pivotal driver of ES oncogenesis, suggesting a suppressor effect.

Taken together, all of these findings contribute to the development of new therapeutic targets for ES management as well as to the better understanding of the physio-pathological mechanisms of ES.

### 4.3. Role of miRNAs in Chondrosarcoma

Chondrosarcoma is the third most common primary bone malignancy after myeloma and osteosarcoma. CS grows slowly, and rarely metastasizes; it is characterized by the formation of hyaline cartilaginous neoplastic tissue. CS accounts for approximately three new cases per 10^6^ population per year. CS is characterized by genomic instability and multiple genomic aberrations, which include TP53 gene, IDH, and PTEN mutations [113].

The prognosis for the majority of patients with CS is favorable and correlates with histologic grade and adequate surgical margins. However, CS exhibits resistance to both chemotherapy and radiation treatment [114].

Various studies have demonstrated that miRNAs are intimately involved in CS oncogenesis. In particular, great attention has been given to the role of miR-100 in CS. miR-100 is downregulated in CS; it is considered a tumor suppressor that targets and inhibits the mammalian target of rapamycin (mTOR) signaling pathway, which is generally involved in tumor growth and metastasis. Overexpression of miR-100 complementary pairs to the 3′ untranslated region (UTR) of mTOR results in sensitization of cisplatin-resistant cells to cisplatin [115]. Thus, overexpression of miR-100 might be exploited as a therapeutic strategy along with cisplatin-based combined chemotherapy for the treatment of clinical chondrosarcoma patients [116].

Similarly, miR-30a decreases tumor proliferation, migration, and invasion through targeting of oncogenic SRY-related HMG box 4 (SOX4) [117]. In contrast, miR-181a is considered a CS oncogene, as it is more expressed in high-grade CS; it is up-regulated by hypoxia, increasing vascular endothelial growth factor (VEGF) expression by targeting regulator of G-protein signaling 16 (RGS16). Moreover, miR-181a is able to negatively modulate the CXC chemokine receptor 4 (CXCR4) signaling pathway [106].

These studies demonstrate the complex interplay of miRNA in CS oncogenesis, suggesting new therapeutic strategies for CS management.

## 5. miRNAs as Diagnostic and Predictive Biomarkers in Bone Tumors

miRNAs are gaining significant interest in the research world thanks to their application as both therapeutic targets and as diagnostic biomarkers in many human diseases, including tumors [80,118].

It has been demonstrated that miRNAs circulating in such body fluids as plasma, serum, etc., are relatively stable and resistant to RNase degradation, probably owing to their small size [119]. Studies have demonstrated that miRNA expression profiles reflect tumor origin, stage, and other pathological variables; this makes them attractive biomarker candidates for cancer diagnosis and monitoring of patients’ responses to therapy. In contrast to standard tissue biopsies, sampling of biofluids for miRNAs is quick, minimally invasive, and painless [119]. Individual miRNAs may be detected with a resolution down to single nucleotide [119]; furthermore, there are indications that the power to distinguish normal from cancer samples may be higher than with traditional biomarkers such as proteins and mRNAs [119]. miRNAs are readily quantifiable in the plasma and blood of patients with bone tumors through different molecular techniques such as quantitative reverse transcriptase-polymerase chain reaction (RT-PCR), in situ hybridization techniques, and hybridization-based microarray platforms [120]; moreover, miRNAs can be detected in routinely prepared formalin-fixed paraffin-embedded (FFPE) materials.

Once detected, miRNA profiles allow for distinguishing between normal and cancerous tissue as well as identifying different tumor subtypes and even specific oncogenic abnormalities [80,115].

An important study conducted by Zou et al. analyzed miR-19a expression and its clinical implications in patients with OS [121]. In this paper, the results demonstrated that high miR-19a expression levels are correlated with large tumor size, advanced clinical stage, positive distant metastasis, and poor response to chemotherapy, suggesting that miR-19a could represent a novel prognostic marker for OS patients [121].

Another important miRNA used as biomarker for diagnosis and prognosis in patients with OS is miR-191 [122]. A study conducted by Wang et al. [122] revealed that increased expression of miR-191 may be associated with aggressive tumor progression, metastasis, and adverse outcomes; therefore, serum miR-191 quantification may be a promising biomarker for the diagnosis and prognosis in osteosarcoma [122].

Thus, differentially-expressed miRNAs in the serum and bone tissues of OS, ES, and CS patients provide a new basis for early diagnosis, prognosis, and targeted therapy of these bone tumors. In addition to their use as predictive biomarkers, the correlation between miRNA expression and response to specific therapies suggests their promising potential as therapeutic adjuvants [80,82,83,84,85,86,87,88]. miRNAs could represent a new class of diagnostic and prognostic biomarkers in bone tumors, especially OS, ES, and CS.

## 6. Conclusions

Despite countless scientific advances, current therapies and diagnostic procedures for OS, ES, and CS have not yet proven very satisfactory. Thus, continuous research is needed in order to identify new molecular targets and new therapies for OS, ES, and CS and to improve outcomes and quality of life for patients. Thanks to their direct involvement in biological processes and in human diseases such as tumors, miRNAs could reveal useful treatments and early diagnosis techniques for bone tumors such as OS, ES and CS. It is well-demonstrated that miRNAs can be valid biomarkers for cancer diagnosis and prognosis. To date, however, the available evidence on miRNAs as diagnostic and prognostic biomarkers for bone tumors is only in the early stages. The present state of knowledge on miRNAs suggests that they are able to regulate osteoblast differentiation and bone formation and that their dysregulation induces bone degeneration and bone-related diseases, thus conferring relevant value for both medicine and basic research. Therefore, further studies are needed in order to better understand their role and function in bone tumors in a way that could be useful for the development of new biomarkers for the treatment and early diagnosis of OS, ES, and CS.

## 7. Future Perspectives

In the field of oncology research, miRNAs represent very interesting molecules to study for the development of new therapeutic approaches to counteract bone tumor progression. Advances in molecular techniques have contributed to better understanding of the molecular mechanisms involved in bone tumors. Thanks to their multiple properties, miRNAs could represent alternatives therapeutic treatment vectors for the management of bone tumors such as OS, ES, and CS, either alone or in association with common chemotherapy drugs, thereby improving patients’ quality of life.

## Figures and Tables

**Figure 1 ijms-23-02348-f001:**
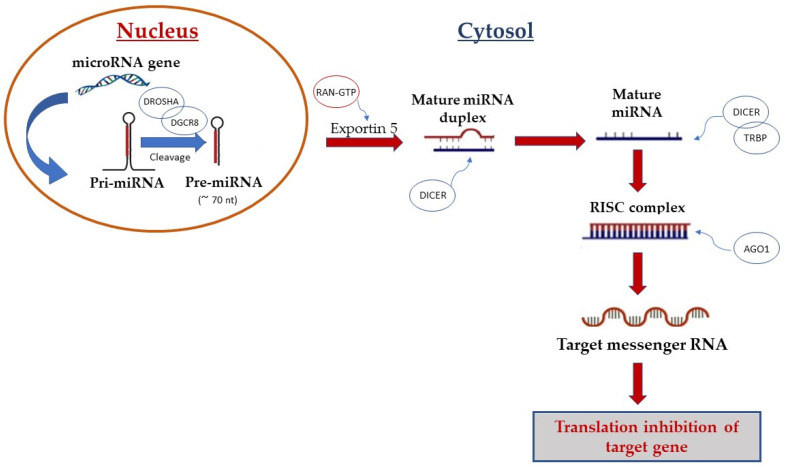
Biogenesis of microRNA. Canonical miRNA biogenesis begins with the generation of the pri-miRNA transcript by RNA polymerase II. The microprocessor complex, comprised of Drosha and DGCR8, cleaves the pri-miRNA to yield the precursor-miRNA (pre-miRNA). The pre-miRNA is transferred from the nucleus into the cytoplasm by Exportin-5 protein. In the cytoplasm, pre-miRNA is processed by Dicer enzymes to produce the mature miRNA duplex. Finally, single-stranded mature miRNA is loaded into the Argonaute (AGO) family of proteins to form the miRNA-induced silencing complex (miRISC).

**Figure 2 ijms-23-02348-f002:**
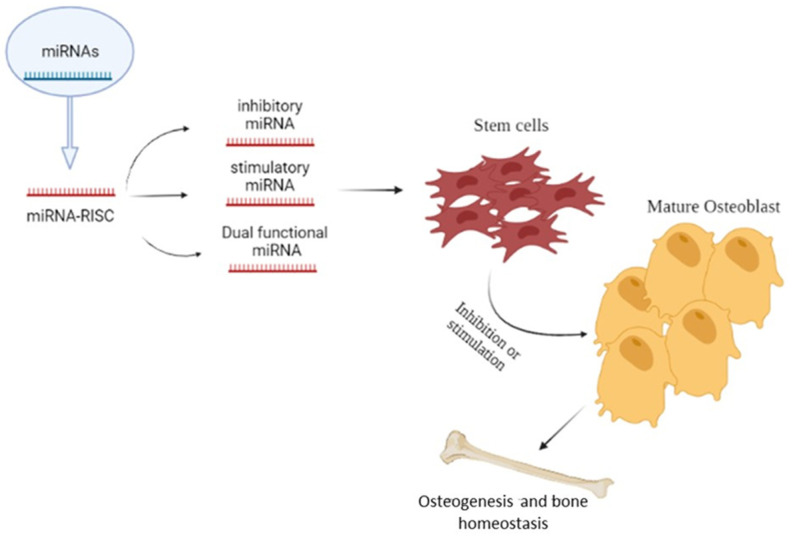
Role of microRNAs in osteogenesis and bone homeostasis; the Figure illustrates the possible inhibitory or stimulatory role of microRNAs in osteogenesis and bone homeostasis.

**Figure 3 ijms-23-02348-f003:**
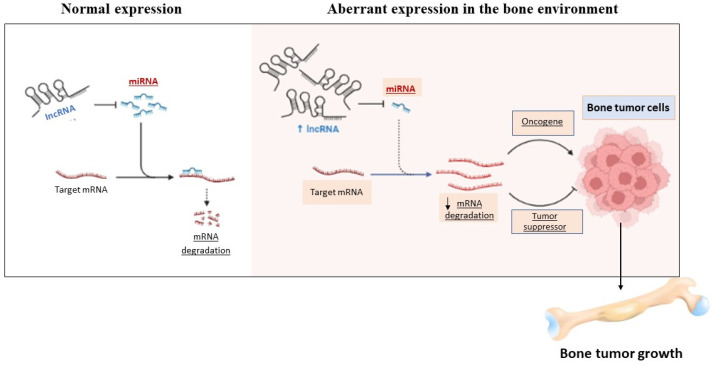
The figure shows the controversial role of miRNAs in the bone environment.

**Figure 4 ijms-23-02348-f004:**
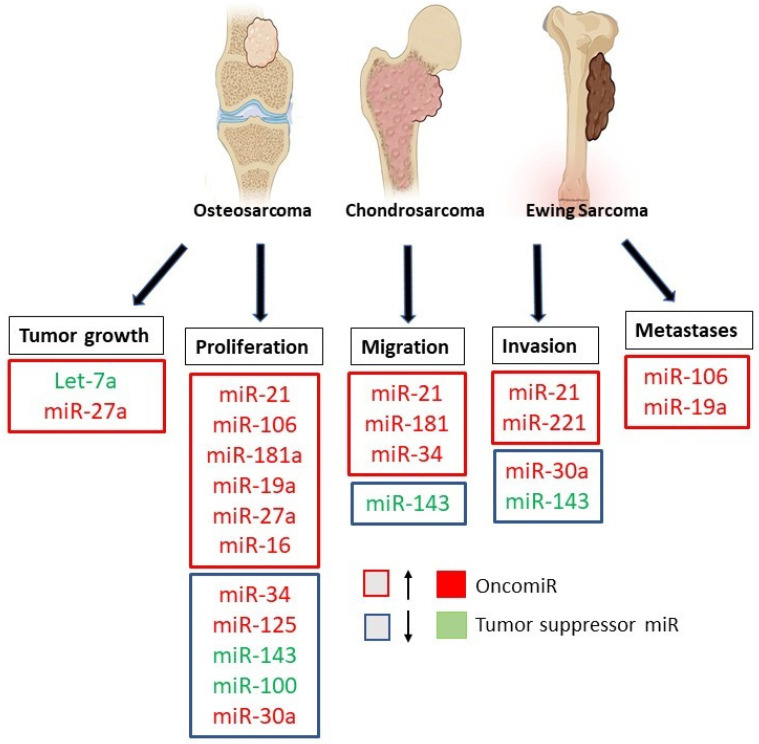
Role of microRNAs in Osteosarcoma, Ewing’s Sarcoma, and Chondrosarcoma. The figure summarizes the down-regulated and up-regulated miRNAs involved in the progression, invasion, and metastasis of bone tumors.

**Table 2 ijms-23-02348-t002:** List of microRNAs involved in Osteosarcoma, Chondrosarcoma, and Ewing’s Sarcoma.

Osteosarcoma	Chondrosarcoma	Ewing’s Sarcoma
miRNA-21	miRNA-21	miRNA-21
miRNA-106	miRNA-181	miRNA-106
miRNA-181a	miRNA-143	miRNA-19a
miRNA-16	miRNA-34	miRNA-30a
miRNA-34	miRNA-145	miRNA-143
miRNA-100	miRNA-221	miRNA-22

## Data Availability

The authors declare that all data and materials supporting the findings of this study are available within the article. The data that support the findings of this study are available from the corresponding author upon reasonable request.

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
