# Peer review of "The Biological Function of MicroRNAs in Bone Tumors"

_ijms, 2022, doi:10.3390/ijms23042348_

Round 1
Reviewer 1 Report
The paper is well-written and provides sufficient coverage of the topic. The majority of the references were published before 2019. The authors need to include more recent studies.
Author Response
#Reviewer 1
The paper is well-written and provides sufficient coverage of the topic. The majority of the references were published before 2019. The authors need to include more recent studies.
As suggested by the reviewer, the authors included more recent studies in the text (doi:10.1002/jev2.12056; doi:10.1186/s12935-021-01780-8, etc).
Reviewer 2 Report
In this article the author want to describe the role of miRNAs in bone cancer. The article has some things that need to be considered:
- In figure 1 authors need to complete the all the important parts of the microRNA biogenesis.
- The section 2.1 describes similar things as section 3 so should be included in that part of the article
- Table on should be completed with other miRNAs that are important in bone cancer (eg. mir-101,mir- 22, mir-483 etc).
- In table 1 each miRNA described has more target genes that the authors indicated, so please add al the target genes and explain why you selected only those genes.
- A table like Table 1 should be made for each type of bone cancer and added in that specific section.
- in the section dedicated to miRNAs as diagnosis or predictive biomarkes the authors should describe more in detail this part and also give some specific examples of miRNAs.
Author Response
#Reviewer 2
In this article the author want to describe the role of miRNAs in bone cancer. The article has some things that need to be considered:
- In figure 1 authors need to complete the all the important parts of the microRNA biogenesis.
As suggested by the reviewer, the authors better described the microRNA biogenesis in the figure 1.
- The section 2.1 describes similar things as section 3 so should be included in that part of the article
As suggested by the reviewer, the authors included the paragraph 2.1 in the section entitled “Relationship between miRNAs and cancers: focus on bone tumors “.
- Table on should be completed with other miRNAs that are important in bone cancer (eg. mir-101, mir-22, mir-483 etc).
As suggested by the reviewer, the authors added other miRNAs involved in bone tumor in Table 1.
- In table 1 each miRNA described has more target genes that the authors indicated, so please add al the target genes and explain why you selected only those genes.
As suggested by the reviewer, the authors added other target genes in Table 1, selecting the most involved in bone tumor, as mentioned in the figure legend.
- A table like Table 1 should be made for each type of bone cancer and added in that specific section.
As suggested by the reviewer, the authors created a new table indicating miRNAs involved in each type of bone tumor (new Table 2).
- In the section dedicated to miRNAs as diagnosis or predictive biomarkes the authors should describe more in detail this part and also give some specific examples of miRNAs.
As suggested by the reviewer, the authors better described the use of miRNAs as predictive biomarkers and provided some examples as miR-19a and miR-191 in the paragraph entitled “MiRNAs as diagnostic and predictive biomarkers in bone tumors”.
Reviewer 3 Report
The review article titled "The biological function of microRNAs in bone tumors", authored by Sarah Adriana Scuderi and colleagues, provides comprehensive evidence on the multifaceted roles of miRNAs in bone malignancies. I have the following Comments and Suggestions for Authors: 1) In general the manuscript provides multiple lines of evidence on the roles of miRNAs in bone cancer. However, the text suffers from several grammatical and structural errors. For example, in Figure 1 legend (Line 109), "Biogenenesis of microRNAs" instead of "Biogenenesis of microRNAs" etc. There are also several spelling errors, such as in Figure 1 (between lines 108-109) "messenger" instead of "messanger". The authors should consider revising, in order to provide an easy to follow text. 2) The term upregulated and down-regulated miRNA in Figure 2 may be misinterpreted as tumor promoting and tumor-suppressor miRNA. For example, Let-7a family is a known tumor-suppressor miR, that is uperegulated in osteosarcoma and the missing information of functional role, may confuse the reader. I would suggest the authors add a colorer text in each miR (for example Green for tumor suppressor miRs and red for oncomiRs) in order to identify the individual impact of each miR in bone tumors. 1)Author Response
#Reviewer 3
The review article titled "The biological function of microRNAs in bone tumors", authored by Sarah Adriana Scuderi and colleagues, provides comprehensive evidence on the multifaceted roles of miRNAs in bone malignancies. I have the following Comments and Suggestions for Authors:
- In general, the manuscript provides multiple lines of evidence on the roles of miRNAs in bone cancer. However, the text suffers from several grammatical and structural errors. For example, in Figure 1 legend (Line 109), "Biogenenesis of microRNAs" instead of "Biogenenesis of microRNAs" etc. There are also several spelling errors, such as in Figure 1 (between lines 108-109) "messenger" instead of "messanger". The authors should consider revising, in order to provide an easy-to-follow text.
As suggested by the reviewer, the authors grammatically revised and improved the manuscript.
2) The term upregulated and down-regulated miRNA in Figure 2 may be misinterpreted as tumor promoting and tumor-suppressor miRNA. For example, Let-7a family is a known tumor-suppressor miR, that is uperegulated in osteosarcoma and the missing information of functional role, may confuse the reader. I would suggest the authors add a colorer text in each miR (for example Green for tumor suppressor miRs and red for oncomiRs) in order to identify the individual impact of each miR in bone tumors.
As suggested by the reviewer, the authors added colored text green for tumor suppressor miRs and red color for oncomiRs in the figure (New Figure 4) to distinguish the impact of each miRNA in bone tumor.
Round 2
Reviewer 2 Report
The author have adresse my comments, but still there are some parts that need to be added. In my comment related to Table 1 I specify only some miRNAs , but the author added only those that I specify, they should have added more.
Author Response
#Reviewer 2
The authors have adresse my comments, but still there are some parts that need to be added. In my comment related to Table 1 I specify only some miRNAs, but the author added only those that I specify, they should have added more.
The authors apologyze for the misunderstanding. As suggested by the reviewer, the authors added other miRNAs in the Table 1.
Reviewer 3 Report
The revised manuscript has been substantially improved.
The authors had taken into consideration my suggestion and succefully answered my comments.Therefore I endorse publication of the current draft.
Author Response
#Reviewer 3
The revised manuscript has been substantially improved.
The authors had taken into consideration my suggestion and succefully answered my comments. Therefore, I endorse publication of the current draft.
Thank you for the comments.